# Fault Diagnosis of Rotary Parts of a Heavy-Duty Horizontal Lathe Based on Wavelet Packet Transform and Support Vector Machine

**DOI:** 10.3390/s19194069

**Published:** 2019-09-20

**Authors:** Hongyu Jin, Avitus Titus, Yulong Liu, Yang Wang, Zhenyu Han

**Affiliations:** 1School of Mechatronics Engineering, Harbin Institute of Technology, Harbin 150001, China; jinhy@hit.edu.cn (H.J.); avitustitus@sua.ac.tz (A.T.); liuyulong@163.com (Y.L.); wyyh@hit.edu.cn (Y.W.); 2Department of Engineering Sciences and Technology, Sokoine University of Agriculture, Morogoro 255, Tanzania

**Keywords:** fault diagnosis, wavelet packet transform, power spectrum analysis, pattern recognition, support vector machine

## Abstract

The spindle box is responsible for power transmission, supporting the rotating parts and ensuring the rotary accuracy of the workpiece in the heavy-duty machine tool. Its assembly quality is crucial to ensure the reliable power supply and stable operation of the machine tool in the process of large load and cutting force. Therefore, accurate diagnosis of assembly faults is of great significance for improving assembly efficiency and ensuring outgoing quality. In this paper, the common fault types and characteristics of the spindle box of heavy horizontal lathe are analyzed first, and original vibration signals of various fault types are collected. The wavelet packet is used to decompose the signal into different frequency bands and reconstruct the nodes in the frequency band where the characteristic frequency points are located. Then, the power spectrum analysis is carried out on the reconstructed signal, so that the fault features in the signal can be clearly expressed. The structure of the feature vector used for fault diagnosis is analyzed and the feature vector is extracted from the collected signals. Finally, the intelligent pattern recognition method based on support vector machine is used to classify the fault types. The results show that the method proposed in this paper can quickly and accurately judge the fault types.

## 1. Introduction

Heavy-duty machine tools are characterized by the ability to process large and heavy components. This working characteristic requires that the machine must have high safety, stability and reliability. Working load and the inertia of the workpiece are very big under this working condition; therefore, once the machine tool component failure causes the sudden stop of the machine, this damages the rotary parts in the spindle box of machine tool [1]. Thus, there has been widespread concern regarding a guarantee of the normal working state of machine tools and the early diagnosis of faults. Faults in heavy duty machine tool components usually occur in rotating parts because, under this kind of heavy load condition, the tiny flaw of the rotating part will expand rapidly. This can cause parts to fail and even have a major impact on other parts. Therefore, early fault diagnosis and determination is of great significance for the use and maintenance of heavy duty machine tools [2].

Methods used for gear box analysis include spectral analysis, cepstrum analysis, demodulation analysis, statistical analysis, and envelope analysis. The appearance of fast Fourier transform (FFT) [3] brings the fault diagnosis method based on Fourier analysis to a new stage. However, people gradually find that the traditional FFT method cannot effectively display the transient signal of fault characteristics. Although envelope spectrum analysis can show fault characteristics, it cannot effectively process unsteady state signals [4]. Some methods have been developed and applied to deal with such unsteady problems. One of them is the short-time Fourier transform (STFT) [5]. This method analyzes the signal at equal intervals. However, since its time-domain resolution cannot be changed, it has problems in analyzing signals with high time-domain resolution. Wigner–Ville distribution (WVD) [6] can be used to analyze specific fault signals with instantaneous energy due to its high resolution in time domain and frequency domain. However, the defect of this method is that it will generate cross terms. The empirical mode decomposition (EMD) [7] method and the Hilbert spectrum transform method are also used to analyze the nonlinear and unstable signals caused by the local faults of the gearbox. Wu used the EMD method to process the de-noising signal and decomposed it into IMF components adaptively, and the corresponding noise and false components were eliminated by the correlation number criterion. However, the disadvantage of the EMD method is that it consumes a lot of computing resources to decompose the computing steps. In recent years, wavelet transform has been successfully applied to the processing of unsteady signals and the extraction of embedded features [8,9]. McFadden points out that continuous wavelet transform is able to achieve a series of resolutions when analyzing signals due to the use of variable scaling functions, and it has been proved in experiments that it has advantages in analyzing the abrupt wavelet transform in gear vibration [10]. Purushotham adopts the method of discrete wavelet transform to decompose the vibration signals of normal bearing and bearing with complex faults respectively, and compares the extracted characteristic data [11]. Although the wavelet analysis method has been adopted by many scholars, there is no accurate conclusion about which wavelet basis function is the most suitable to analyze the data.

The working state of gear or bearing can be identified by analyzing the characteristic parameters of fault. From the current research, the development trend is that the machine can recognize and judge the fault type by itself. Many scholars also focus on the application of artificial intelligence, machine learning and other technologies to achieve automatic fault identification. Among all artificial intelligence methods, artificial neural networks (ANNs) are the most widely used. Rafiee [12] used the multi-layer perceptron neural network to deal with the identification of gear faults and bearing defects, and found in the application that feature vectors play a crucial role in the identification effect of the trained neural network. In addition to ANNs, some new methods such as support vector machine (SVM), relevance vector machine (RVM), and hidden Markov model (HMM) are proposed for fault diagnosis and classification. Samanta compared the recognition effect of ANNs and SVM for bearing faults under different loads and speeds, and SVM was better than ANNs in terms of diagnosis accuracy and training rate. In addition, the author also used a genetic algorithm (GA) [13] to optimize the structural characteristics of ANNs, such as the number of layers and nodes, and the kernel function of SVM. Both of them performed better after optimization than before optimization, which also provided a good guidance for better application of intelligent diagnostic tools. The advantage of RVM is that it gives up the decision function, which can obtain better relaxation characteristics than SVM [14]. When processing the same data, the RVM method with a linear feature extraction feature can give more successful judgment conclusion than SVM. Purushotham introduced an HMM model that is successfully used in speech recognition. Based on the signal data measured in the experiment, the components with unknown faults are diagnosed and good judgment results are obtained. Ge also studied the HMM method and pointed out that it could classify shock vibration. However, the judgment effect largely depends on the quality of the training vector, so the autoregressive model (AR) is introduced to assist the extraction of feature vectors [15]. Although the above methods can achieve fault classification, how to select a method that can guarantee both the robustness of classifier and the recognition accuracy still needs to be studied.

In view of the shortcomings of existing literature, this paper provides a new model by combining frequency domain analysis and pattern recognition to realize the intelligent fault diagnosis of a machine tool spindle box. To effectively realize fault feature extraction, the wavelet packet is used to decompose the signal into different frequency bands and the power spectrum analysis is carried out on the reconstructed signal. The structure of the feature vector used for fault diagnosis is established based on the detected vibration signals. The intelligent pattern recognition model based on a support vector machine is used to identify the faults in the spindle box of a heavy-duty horizontal lathe.

## 2. Fault Detection and Classification of Rotary Parts

### 2.1. Fault Classification of Spindle Box

According to the different parts of the gear box, the failure can be divided into gear failure, shaft failure, and bearing failure. Among them, gear faults mainly include tooth surface wear, tooth surface abrasion, tooth surface contact fatigue and tooth fracture. Bearing faults can be divided into inner ring fault, rolling body fault, cage fault and outer ring fault according to the location. Shaft failures include shaft bending and shaft imbalance and misalignment. In addition, as an assembly box, the main assembly faults include too large tooth backlash and loose assembly.

As it is a new machine tool to be delivered, there is no failure such as tooth wear and bearing failure. For the spindle box of heavy duty horizontal lathe, the faults mainly include: (1) misalignment existing between motor shaft and axis; (2) misalignment existing between transmission gears in the box; (3) loose assembly of the transmission shaft in the gearbox; and (4) the backlash of two pairs of meshing gears being too large.

#### 2.1.1. Coaxiality out of Tolerance of The Connecting Shaft

The motor of the machine tool is connected to shaft I through the shaft coupling. The installation of this transmission structure will be accompanied by the connection shaft coaxiality out-of-tolerance, which is usually manifested as axis parallel misalignment and angle misalignment coexistence, as shown in Figure 1a. In the case of coaxiality error between connecting shafts, the radial vibration frequency of slave shaft is twice that of master shaft, and with the increase of misalignment angle “α”. In addition, when the misalignment phenomenon exists, it is generally the axis parallel misalignment and the angle misalignment appearing at the same time. The radial vibration of slave shaft appears to have double frequency, and the more serious the misalignment, the greater the proportion and amplitude of double frequency. In addition, in general, the amplitude of double frequency is higher than the fundamental frequency. The general form of its spectrum diagram is shown in Figure 1b, where *f* is the frequency of master shaft.

#### 2.1.2. Misalignment of Gears

In the process of machining and assembly of gear box body, it often occurs that the position accuracy between the holes does not meet the requirements due to the machining error of the box body, which leads to the misalignment of the two axes equipped with meshing gears after assembly. In addition, some gears in the spindle box of the heavy horizontal lathe are connected to the shaft by spline. Because the coaxiality of the shaft is out of tolerance and the coaxiality of the gear is out of tolerance, the gear and the shaft will be misaligned. These two phenomena will lead to the uneven load of the gear in the direction of the width of the tooth, the stability of power transmission decreases, and the load on the local area of the tooth surface increases. The assembly error diagram of different axes is shown in Figure 2a.

The misaligned gear will produce meshing frequency vibration with side frequency band at meshing frequency. Moreover, there are harmonics of meshing frequency, and the peak value of harmonics at double and triple meshing frequency is very high, even higher than the peak value of meshing fundamental frequency. In addition, the modulation phenomenon appears around the peak value of each order harmonic frequency, and the modulation frequency is consistent with the rotation frequency of the shaft where the gear is located. The frequency spectrum of gear misalignment is shown in Figure 2b.

#### 2.1.3. Excessive Backlash of Gear Tooth

In the assembly of the spindle box, the tooth backlash of the meshing wheel teeth will be too large after assembly. Because the tooth backlash will affect the load change of the gear transmission, the vibration characteristic of the gear system will change and the peak frequency will change. When the tooth backlash is too large, the fractional multiple harmonics of meshing fundamental frequency will appear in the spectrum diagram in addition to the high harmonics of meshing fundamental frequency. The spectrum diagram is shown in Figure 3.

#### 2.1.4. Loose Mounting of Bearing Seat

During the installation of the drive shaft, the bearing supporting the shaft is mounted on the bearing seat in the box body. In the process of installation, there may be loose installation of bearing pedestal due to loose installation of connecting bolts of bearing pedestal. Once it appears, it will cause strong vibration and noise when the box works. Although the frequency of such faults is lower in practice than those mentioned above, it still needs to be accurately diagnosed and eliminated in view of its serious interference to normal working conditions.

In the case of installation looseness, in addition to the basic frequency of axis rotation *f*, there will be high order harmonic frequency of 2*f*, 3*f*, 4*f* with equal axis rotation frequency, and fractional frequency components of basic frequency such as *f*/2, 2*f*/3, and *f*/3 in the frequency spectrum. In addition, the amplitude of these frequencies is significantly increased compared with the normal operation, but the rotational fundamental frequency is still the dominant one. Therefore, the amplitude of each characteristic frequency should be comprehensively analyzed when diagnosing this kind of fault. The spectrum diagram is shown in Figure 4.

### 2.2. Fault Signal Acquisition

This paper mainly analyzes the vibration signal accompanied by the fault, so it is necessary to collect the vibration signal of the gearbox in the time domain. Sensors suitable for the acquisition system of this paper should be able to pick up vibration signals caused by assembly faults such as misalignment of shaft and loose installation. In addition, in order to better analyze the signal in the time domain and frequency domain, the shaft speed should also be measured.

According to the corresponding relationship between acceleration and force, the piezoelectric effect can be used to determine the corresponding relationship between acceleration and voltage signal. Moreover, the piezoelectric effect is characterized by good linearity and high sensitivity, and its frequency measurement range can be from 0.0001 Hz to 1 MHz. Therefore, the piezoelectric acceleration sensor has the advantages of high sensitivity and spectral bandwidth. In addition, most of the installation of the acceleration sensor only needs to be fitted with the measured object. The fixing method includes bonding it with the surface to be tested by adhesive, or fixing it by the external force of magnetic seat. In this paper, an acceleration sensor (PCB type 356A16) is adopted, as shown in Figure 5b. The sensitivity is 100 mV/g. The corresponding frequency range is 0.5 to 5000 Hz, the measuring range is 100 g, and the weight is 7.4 g. In the application, the magnetic seat can be bonded on the end cover of the bearing seat.

In order to better filter out the fault information contained in the vibration signal, it is necessary to know the actual rotation speed of the shaft during the test. In this way, the possible failure frequency of each drive shaft and the parts on the shaft can be calculated. By combining this information with the processed original vibration signal, the fault of the spindle box can be diagnosed. Since the number of gear teeth on each drive shaft of the gearbox can be obtained by consulting the design manual of the corresponding machine tool, it is only necessary to understand the rotation speed of the power input motor.

In this paper, rotation speed tester (mvp-2c) is used to measure the shaft speed. During the shaft rotation, the optical signal will be reflected by the reflector when it irradiates the reflector, and the appearance and disappearance of such reflection will trigger the photoelectric sensor to generate pulse signal. By analyzing the pulse signal, the instrument can output the value of rotation speed and achieve the test purpose. This method is simple, fast and suitable for complex environment operation. The instrument and test environment are shown in Figure 5c.

In this paper, the LMS test instrument is used for data collection, which can realize data collection in continuous time with different sampling frequencies under multiple channels. In addition, its parameter setting and adjustment are implemented by software installed on the PC. The tester is easy to operate and can adjust the sampling frequency from 0 to 10 kHz. The hardware configuration and software interface of the LMS signal acquisition system are shown in Figure 5d,e.

The signal line of the sensor is connected to the front acquisition box with a multi-channel interface. There is a module for amplifying the original signal in the pre-acquisition cabinet, which can ensure the signal strength. The prE–Signal acquisition box is connected with the computer, and the parameter setting, start and end of the acquisition are controlled by the computer. The three curves in the acquisition software interface correspond to the *x*-, *y*- and *z*-direction input signals of the vibration sensor, respectively. The collection time, sampling frequency and other parameters can be set in the toolbar on the right side of the software. After the sampling starts, the system continuously collects the signals from the sensor. After the set acquisition time arrives, the sampling stops. Set parameters, sensor information and collected data information are recorded and stored in the same file after the completion of sampling. These files provide sample data for subsequent data processing.

The vibration sensor is attached to the bearing seat or the end cover of the bearing seat in the experiment. In this way, the signal source is relatively close to the transmission medium, and the exchange surface of the transmission medium is relatively small, which can ensure the signal strength of the characteristic information: signal sampling in order for the motor, axis I, axis II, and axis III. The layout of the sensor is shown in Figure 6.

## 3. Wavelet Packet Transform Based Fault Feature Extraction

### 3.1. Determination of Wavelet Basis Function

Because wavelet transform can map the signal to time–frequency domain, and a series of wavelet basis functions are used to represent the signal, the hidden features in the signal can be effectively displayed. The assembly failure of spindle box studied in this paper is a part of rotating machinery failure. In order to better extract the characteristics of the collected samples, the wavelet analysis method is also used to process the fault signal.

The basic method of wavelet transform is to take the inner product of the basic wavelet function with the signal x(t) to be analyzed at different scales *a*. The basic expression of continuous wavelet transform is shown as Equation (Equation 1):(1)$$WT_{x}\left(a,b\right)=\langle x,\psi_{a,b}\rangle =\frac{1}{\sqrt{a}}\int_{-\infty }^{+\infty }x\left(t\right)\bar{\psi }\left(\frac{t-b}{a}\right)dt,$$
where $WT_{x}\left(a,b\right)$ is Wavelet coefficients, *a* is scale that represents the frequency parameter, and *b* is time or space position parameters.

For continuous wavelet transform, scale *a*, time *t* and offset *b* are continuous. When using the computer to realize the calculation of wavelet transform, they need to be discrete processing, which is the discrete wavelet transform. Binary discretization of scale *a* and offset *m* is usually carried out, and the corresponding binary wavelet is shown in Equation (Equation 2):(2)$$\psi_{j,k}\left(t\right)=2^{-\frac{j}{2}}\psi \left(2^{-j}t-k\right).$$

In the calculation procedure of binary discrete wavelet transform, only the approximate part (low-frequency sub-band) of the signal is divided by exponential interval from scale 2, and the detailed part (high-frequency sub-band) is not processed. Therefore, the resolution of the high frequency part is poor. In order to process and analyze the high frequency part, Coifman, Meyer and Wickerhauser put forward the concept of wavelet packet decomposition. Wavelet packet decomposition divides the frequency band into several levels, so as to further decompose the high frequency part and realize more detailed analysis of the signal. The decomposition structure obtained by decomposing signal *S* into three-layer wavelet packet transform is shown in Figure 7.

It can be seen from the analysis structure diagram that wavelet packet analysis can decompose the original signal into each frequency band, so that the frequency band matching the concerned frequency can be selected for analysis, and the time-frequency resolution of signal analysis can be improved.

The fault characteristic frequency studied in this paper is not limited to the rotation frequency or meshing base frequency of a rotating part. In order to realize fault diagnosis, their high harmonic frequencies are also analyzed. Therefore, the wavelet packet transform method can be used as a filter. A node only reflects the information related to the frequency range of the node but excludes the interference of other frequency components. In contrast, discrete wavelet transform usually ignores some high frequency information. In this paper, the advantages and disadvantages of wavelet transform and wavelet packet transform in signal representation will be compared quantitatively.

The evaluation criteria of wavelet transform method are mainly as follows:

(a) The energy content of the signal

The energy content of a signal is a parameter directly related to the characteristics of the signal, so it is usually used to describe the signal. The energy contained in the signal x(t) can be calculated by the wavelet coefficient, as expressed by Equation (Equation 3) [16]:(3)$$E_{energy}=\sum_{s}\sum_{i}{\left|W(s,i)\right|}^{2}.$$

If a major frequency component of a signal corresponds to a specific scale, the wavelet coefficients in that scale will have a relatively large amplitude when the major frequency occurs. This allows the energy associated with a particular frequency component to be extracted from the signal. Therefore, the energy content can be used as a criterion to evaluate the applicability of wavelet basis. For the same signal, the larger the calculated value of the energy content, the more effective and appropriate processing method is considered to show the characteristics of the signal, which is the maximum energy content standard.

(b) Shannon entropy of the signal

The wavelet coefficients are obtained by wavelet transform. The coefficients reflect the similarity between wavelet and signal. If the coefficient matrix obtained by wavelet transform is regarded as a distribution, the sparser the coefficient matrix is, the higher the similarity between signal and wavelet will be. Shannon entropy can be used as an indicator to evaluate sparsity. Shannon entropy is defined as Equation (Equation 4):(4)$$\left\{\begin{array}{c} E_{entropy}\left(s\right)=-\sum_{i=1}^{N}p_{i}log_{2}p_{i},\\ p_{i}=\frac{{\left|wt(s,i)\right|}^{2}}{E_{energy}\left(s\right)},\sum_{i=1}^{N}p_{i}=1, \end{array}\right.$$
where $p_{i}$ is the energy probability distribution of the wavelet coefficients.

The probability distribution uniformity can be evaluated by calculating the Shannon entropy of the distribution. For example, the distribution is the most uncertain when it has equal probability property. This distribution has the highest Shannon entropy and the lowest sparsity. This evaluation method is applied to the calculation and evaluation of wavelet coefficient matrix. The smaller the Shannon entropy is, the more the corresponding wavelet analysis method matches with the signal. This is also the minimum Shannon entropy standard to evaluate the applicability of wavelet basis function and wavelet analysis method.

(c) The ratio of energy content to Shannon entropy (E–S ratio)

Both maximum energy content standard and minimum Shannon entropy standard can be used to evaluate wavelet analysis. These methods are based on the description of the original signal information by wavelet coefficients. The evaluation standard derived from the combination of these two is called energy–Shannon entropy ratio (E–S ratio), and its definition is shown in Equation (Equation 5). Corresponding to the maximum energy content standard and the minimum Shannon entropy standard, the wavelet transform method with the maximum E–S ratio has the strongest ability to characterize the signal. The above three evaluation criteria will be verified and compared in the analysis of actual signals later:(5)$$R\left(s\right)=\frac{E_{energy}\left(s\right)}{E_{entropy}\left(s\right)}.$$

In different calculation methods of wavelet transform, discrete transform and wavelet packet transform can be implemented by the computer, and they can decompose signals into different frequency scales. Regarding which method is more suitable for the fault diagnosis of spindle box, the above-mentioned criteria (as shown in Equations (3)–(5)) are used for comparison. Using the same wavelet basis function, the methods of discrete wavelet decomposition and wavelet packet transform are compared. By analyzing the advantages and disadvantages of these two methods for decomposition of original signals, Shannon entropy, energy content and E–S ratio under different decomposition modes are calculated, respectively.

The wavelet basis function coif1 in the Coiflet wavelet system is used, and the discrete wavelet transform is applied to carry out 3-layer decomposition of the original signal. For the wavelet packet, the decomposition of the original signal with the scale of 3 is also carried out. In addition, this comparative analysis was carried out for different samples. Data collected on each axis are analyzed and calculated, and the results are shown in Table 1.

By comparing the data values of each row in the table, it can be seen that Shannon entropy value of signal through wavelet packet transformation is lower than that of discrete wavelet transform. The energy value is larger than the energy obtained by the discrete wavelet transform, and the E–S ratio is correspondingly larger than the discrete wavelet transform. It can be verified that wavelet packet decomposition can better show the change rule of the signal, characterizing the characteristics of signal and the information contained in it. In order to demonstrate the universality of the above verification, the wavelet basis function was replaced to process the same data, and the calculation results of the two analysis methods were still compared. Using wavelet basis function sym4, which is different from wavelet basis waveform coif1, the calculation result is similar to that of coif1 wavelet, which proves that wavelet packet analysis can reflect the characteristics of signal more precisely. From the perspective of principle, wavelet packet analysis not only decomposes the low-frequency signal, but also decomposes the high-frequency signal that has not been decomposed in the discrete wavelet transform, so the description of signal features is more appropriate. Therefore, better calculation results can be obtained when various indexes are introduced for evaluation. In addition, some of the characteristic frequencies studied in this research are at the higher harmonic frequency of the meshing frequency, so the high-frequency part of the signal also needs to be paid attention to. Therefore, wavelet packet decomposition can better meet the research needs.

Based on Shannon entropy, energy content and E–S ratio, the wavelet basis suitable for the analysis of spindle box vibration signal is evaluated and selected. Different wavelet basis functions are used for wavelet packet decomposition transformation, and the results are shown in Table 2.

It can be seen from the calculation results that the values of each index obtained after the processing of different wavelet basis functions are different. Among these wavelet families, the calculated values of Bior and R-bior wavelet families are superior to other wavelet families. The results of wavelet functions of different orders in the same family of wavelets are different. The basic rule is that the higher the order, the better the result of each index. In terms of energy content, the maximum value of Bior3.5 was calculated in the analysis of different groups of sample data. In the calculation of Shannon entropy, Bior3.5, Coif4 and Rbio3.5 respectively obtained a primary minimum value. As for the E–S ratio, the maximum value of Bior3.5 is obtained. It can also be seen that the maximum E–S ratio standard can help us select the most suitable wavelet. Bior3.5 was selected for wavelet packet decomposition in this research.

### 3.2. Fault Feature Extraction

The signal collected from the spindle box of the machine tool is analyzed. The spindle box is known to have gear misalignment errors among the shafts. Sampling signal acquisition in shaft II bearing end cover. The tooth transmission relation between machine each axis: shaft I/II is 28/47, shaft II/III is 30/43. The motor speed measured during sampling was 999.6 rpm and the sampling frequency was 8192 Hz. After calculation, the rotation frequency of the test shaft is 9.93 Hz, and the engagement frequency of the gears installed on the shaft is 466.48 Hz and 297.75 Hz. The original time domain waveform of the acquired signal is shown in Figure 8. The period of some faults cannot be directly identified from the information in the figure, and the waveform peak is disordered. Bior3.5 wavelet is used for five-layer wavelet packet decomposition, so the signal is decomposed into 32 frequency bands, and a five-layer binary tree structure is obtained. The last layer is [5 0], [5 1], …, [5 31], a total of 32 nodes, and the corresponding frequency band is 0–128 Hz, 128–256 Hz, …, 3968–4096 Hz. The characteristic frequency associated with the mismeshing fault of the gear is the first, second and third meshing frequency, and the corresponding frequency of conversion modulation, that is, the meshing frequency plus or minus 9.93 Hz.

The frequency points concerned are 297.75 Hz, 595.5 Hz, 893.25 Hz, 466.48 Hz, 932.96 Hz, 1399.44 Hz, and the modulation frequencies around them. These frequency points are located in frequency bands corresponding to nodes [5 2], [5 3], [5 4], [5 6], [5 7], [5 9]. These nodes were reconstructed and Hilbert transformation was performed for spectral analysis, and the waveform obtained was shown in Figure 9. Figure 9a,c,e deal with pairs of nodes [5 2], [5 4] and [5 6]. Gear meshing frequency between shaft II and shaft III is 297.75 Hz, and the second and third order has a larger peak near the frequency doubling, and they all have a modulation of about 9 Hz. From the size of its peak value, it can be seen that the energy at the second and third order frequencies does not decay, and is even greater than the intensity of the meshing fundamental frequency, which is consistent with the fault characteristics. Figure 9b,d,f processing the node [5 3], [5 7], [5 9] represent axial meshing frequency between shaft I and shaft II, the second order and third order meshing frequencies and the corresponding amplitude modulation frequencies. Through the characteristics shown in these processing results, it is proved that this processing method can be used to obtain significant changes related to specific faults in the frequency–power spectrum, and this ability to capture abnormal features can be used to extract features under different faults.

By analyzing the amplitude of signal at certain frequency in signal power spectrum, the fault type can be judged. Therefore, it is very meaningful to obtain the energy values at these characteristic frequencies for fault analysis. Because there are only certain kinds of assembly faults of the spindle box of the heavy horizontal lathe, and the detection of some faults needs to be based on the amplitude analysis of some rotating frequencies. These can be used as the basis of the characteristics of the frequency mainly includes the following: (1) shaft frequency and its harmonic frequency, shaft rotation frequency of 1/2 frequency, 1/3 frequency; (2) meshing frequency and doubling frequency of the gear mounted on the shaft; (3) the gear mesh frequency plus or minus the frequency of the shaft rotation frequency value. With the amplitudes at these frequency points, they can be arranged into a set of eigenvectors to represent the operation of the axis. The feature vectors extracted from known faults can be used to train the intelligent classifier, while the features extracted from unknown fault signals can be used to judge the fault type of the axis.

Considering the fault types to be diagnosed comprehensively, the feature frequency points that should be extracted are summarized in Table 3. Considering that there are two gears meshing on some axes, the information of their frequency points should be extracted. For the side frequency modulation of the meshing frequency, the fault can be diagnosed only by knowing whether the unilateral modulation phenomenon exists. Therefore, a representative value of the two side frequencies can be extracted. For each signal, a 23-dimensional vector is extracted to represent its characteristics.

In order to extract the power spectrum value at the above characteristic points, the variables should be clearly defined, including the number of teeth of each relevant shaft of the machine tool spindle box, motor speed during testing, gear position during sampling, sampling frequency, and which axis the sensor is arranged in during sampling. After calculating the characteristic frequency, the wavelet packet decomposition node corresponding to the frequency band is reconstructed and spectral analyzed, and then the value of the characteristic frequency point is obtained. At this time, in the algorithm, the maximum value in the range of –2 Hz to +2 Hz can be extracted in order to cope with the rotation speed change in the above analysis. As for the meshing edge frequency band information, only one value needs to be extracted, so the value with a large side frequency is selected to extract it. Since the extracted values are likely to be large, they should be normalized to facilitate subsequent analysis and calculation. $E_{ij}$ represents the amplitude of a certain frequency point, and it is the energy at a discrete point *i* representing the characteristic frequency after reconstruction of the node *j*. The total energy $E=\sum_{j}\sum_{i}E_{ij}$. For the extracted 23-dimensional eigenvector *T*= [$T_{1}$, $T_{2}$, …, $T_{23}$], each value corresponds to the value of a certain $E_{ij}$. After normalization, a new vector T’ can be obtained, T’= [$T_{1}$/*E*, $T_{2}/E$, …, $T_{23}/E$], which will be used for the subsequent intelligent fault classification.

## 4. Fault Diagnosis Based on Support Vector Machine

To realize intelligent and automatic fault diagnosis, a corresponding intelligent algorithm is needed. The acquisition of signals, decomposition and transformation of signals and extraction of fault features mentioned above are all preparations for the realization of fault diagnosis. After obtaining the characteristic vectors representing various types of faults, these samples can be used to study the intelligent classification algorithm. The problem of small sample training classification can be solved by means of an SVM method in this research.

The set of spindle box fault classification can be divided by using a linear function, that is, there is a hyperplane to distinguish samples, as shown in Equation (Equation 6) [17]:(6)$$\left\{\begin{array}{c} \omega^{T}x_{i}+b=0,\\ \omega^{T}x_{i}+b\geq 1,y_{i}=1,\\ \omega^{T}x_{i}+b\leq 1,y_{i}=-1, \end{array}\right.$$
where the nearest distance between ($x_{i}$,$y_{i}$) and the classification hyperplane is $\frac{1}{∥\omega ∥}$, and the hyperplane with the largest distance value $\frac{1}{∥\omega ∥}$ is the optimal hyperplane.

Linear SVM is to find the weights and biases, so that, while Equation (Equation 6) can be satisfied, the maximum $\frac{1}{∥\omega ∥}$ can be obtained. Finding the maximum value of $\frac{1}{∥\omega ∥}$ is the same as finding the minimum value of $\frac{1}{2}{∥\omega ∥}^{2}$, which is the expression of Equation (Equation 7):(7)$$\left\{\begin{array}{c} \min\frac{1}{2}{∥\omega ∥}^{2},\\ s.t.y_{i}\left(\omega x_{i}+b\right)\geq 1,i=1,2,\ldots ,l. \end{array}\right.$$

The Lagrange multiplier is introduced to solve the optimization problem:(8)$$L\left(\omega ,b,\alpha \right)=\frac{1}{2}{∥\omega ∥}^{2}-\sum_{i=1}^{l}\alpha_{i}\left[y_{i}\left(\omega x_{i}+b\right)-1\right],$$
where $\alpha_{i}$, *i* = 1, 2, …, *l* is a Lagrange coefficient. Equation (Equation 8) can be simplified to Equation (Equation 9):(9)$$\left\{\begin{array}{c} \max Q\left(\alpha \right)=\sum_{i=1}^{l}\alpha_{i}-\frac{1}{2}\sum_{i=1}^{l}\sum_{j=1}^{l}\alpha_{i}\alpha_{j}y_{i}y_{j}\left(x_{i}x_{j}\right),\\ s.t.\sum_{i=1}^{l}\alpha_{i}y_{i}=0,\alpha_{i}\geq 0. \end{array}\right.$$

Samples corresponding to $\alpha_{i}$≠0 are support vectors. The optimal solution can be obtained:(10)$$\left\{\begin{array}{c} \omega^{*}=\sum_{i=1}^{l}{\alpha_{i}}^{*}x_{i}y_{i},\\ b^{*}=-\frac{1}{2}\omega^{*}\left(x_{r}+x_{s}\right), \end{array}\right.$$
where $x_{r}$ and $x_{s}$ are any support vector in their respective categories, and the discriminant function of the classifier can be obtained as in Equation (Equation 11):(11)$$f\left(x\right)=\sum_{i=1}^{l}{\alpha_{i}}^{*}y_{i}{x_{i}}^{T}x+b^{*}.$$

When SVM [18,19,20] classification is used, complex problems to be classified are transformed into simple quadratic programming problems, which can guarantee the correct and optimal classification in principle [21]. When the SVM method is applied in practice, the training samples needed are the eigenvectors of corresponding faults that have been identified. Considering that in practice the test shaft may have one fault or several faults at the same time, it is therefore necessary to train SVM with the classification ability for various faults. In principle, there are two ways: one is to train SVM that can distinguish multiple categories of samples, and the other is to train multiple SVM that can only make dichotomies. The first method requires a corresponding sample of each fault combination mode for training, which is difficult to do in practice because the fault types are randomly combined and the sample number is very limited. Considering the second approach, it is possible to train an SVM classifier for each fault to be identified. When features need to be classified and recognized, each SVM is used to identify them, so as to judge whether the signals to be diagnosed have faults corresponding to each SVM in turn. In this way, various single faults and combinations of faults can be identified.

When multiple SVM are used for recognition, features need to be divided because information on a particular dimension in a feature represents a particular failure. The faults to be identified are misalignment of two connecting shafts, looseness of bearing seat installation, misalignment of gears and excessive side clearance of meshing gear teeth. The combination of frequency points that can display these faults is shown in Table 4, where *f* is the rotation frequency of the corresponding shaft and $F_{1}$ and $F_{2}$ are the gear engagement frequencies.

After feature extraction of acquired data by the wavelet packet decomposition method, feature vectors are divided according to the requirements of each SVM, and 48 class A and B samples and 75 class C and D samples are finally obtained. The samples in each category are randomly assigned. One part is used to train the respective SVM, and the other part is used to verify the generalization property of the trained SVM.

As for the selection of kernel function, there is no uniform standard and method, but radial basis function (RBF) has been applied successfully in various cases. When training SVM, in order to achieve better learning and classification effects, it is necessary to select the optimal punishment factor c and RBF kernel parameter *g*, and apply the cross validation method in the optimization process. The idea of this method is to divide the original data into *k* groups according to the number of cross validation *k*. After grouping, the $k^{-1}$ group is used for training, while the remaining group is used for validation. Different groups were selected as verification groups, and the calculation process was carried out *k* times. The optimization range of *c* and *g* is set as $2^{-10}$ to $2^{10}$, the iteration step length is 0.5, the number of cross validation is 5, and the optimization target is the cross validation accuracy of classification. After iteration and optimization, the three-dimensional diagram of the relationship between *c* and *g* and classification accuracy obtained is shown in Figure 10, and the optimization results of each classifier are shown in Table 5.

According to the analysis results, the optimal c and g obtained in SVM training for different types of samples are not the same. Therefore, it is necessary to optimize c and g by cross-validation for each SVM. The optimized SVM model is used to classify the test samples and obtain the accuracy of fault type judgment. The summary is shown in Table 6.

It can be seen from the table that, after training, the classification accuracy of class B, C and D classifiers on test samples reaches over 90%, with high classification accuracy. Meanwhile, the spindle box fault diagnosis method based on WPT and SVM is verified. The classification accuracy of class A is relatively low because the number of samples marked as faults is relatively small. However, its classification accuracy can also reach 80%. External validation has been executed to evaluate the developed prediction model. For the four classifiers, 80% of the data are used to train the model and the remaining 20% are for validation. From the validation results, the accuracy of fault classification results was 81.58%, 94.73%, 95%, and 96.67%, respectively. Test results of different studies by using SVM are shown in Table 7. This shows again that the model proposed in this paper has high precision. In the actual classification of signal testing, signal features can be extracted separately, and each SVM is used to diagnose whether the shaft has a specific fault. For shafts with two pairs of meshing gears, it is necessary to use classes C and D to diagnose the two sets of gears twice, respectively, to determine whether gears have faults or not.

## 5. Conclusions

The fault types of gear box assembly of a heavy-duty horizontal lathe and corresponding frequency spectrum characteristics are determined. The signal acquisition system based on an accelerometer is set up and the vibration signal of the spindle box is collected. Based on Shannon entropy theory, the characteristics of different wavelet transform methods and wavelet basis functions are analyzed. The signal scheme of wavelet packet transformation using Bior3.5 wavelet is determined. A method of fault feature extraction is proposed and the correctness of this method is verified by analyzing the signals of known faults. For the faults to be analyzed, the characteristic vector with a dimension of 23 is determined, and the fault samples for fault diagnosis are obtained. A fault classifier based on support vector machine (SVM) is designed and its relevant parameters are optimized. Fault diagnosis is made by using a classifier. The test results show that, among the four types of faults, the SVM can identify three types of faults with an accuracy rate of over 90%, which is suitable for assembly fault diagnosis.

## Figures and Tables

**Figure 1 sensors-19-04069-f001:**
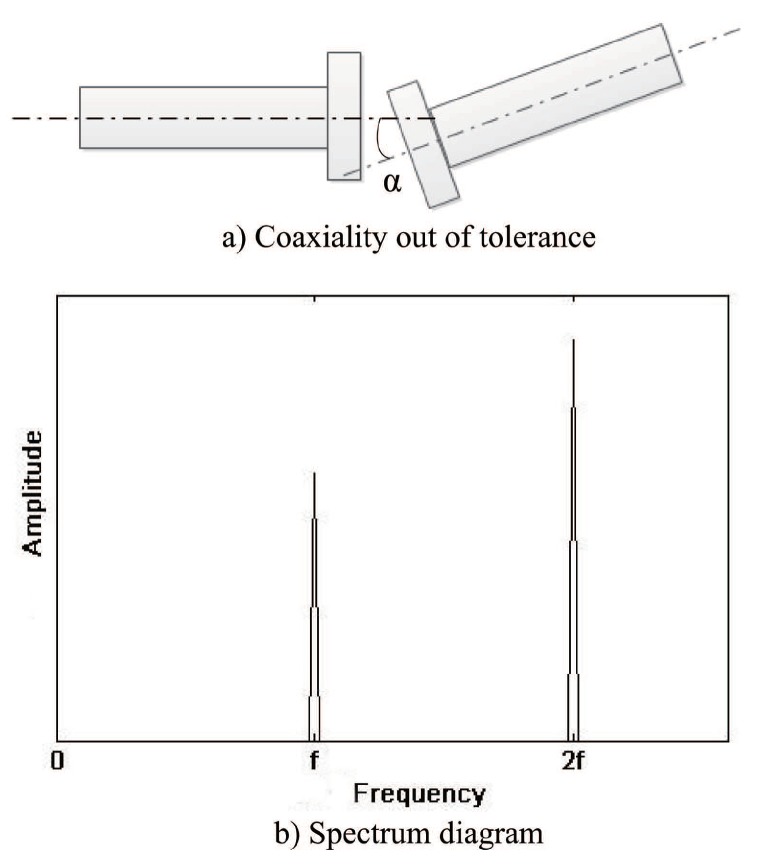
Axis misalignment fault and spectrum diagram. (**a**) Coaxiality out of tolerance. (**b**) Spectrum diagram.

**Figure 2 sensors-19-04069-f002:**
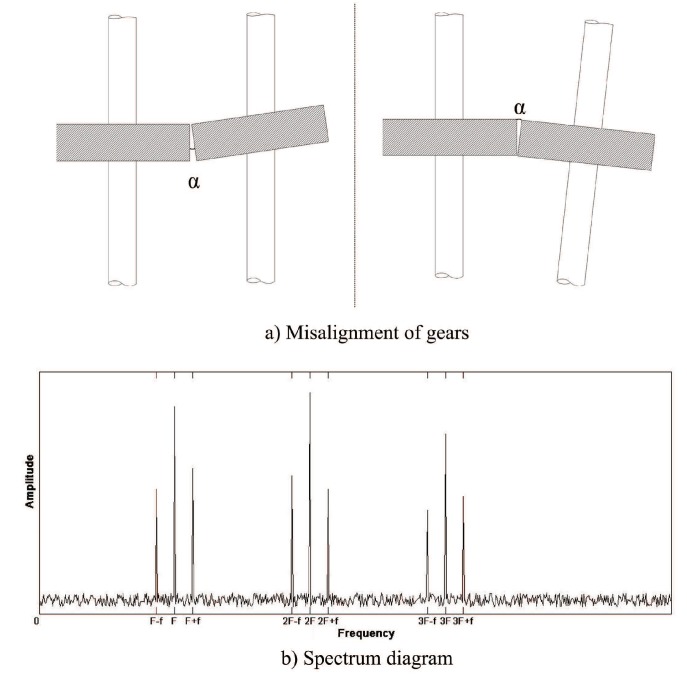
Gear misalignment fault and spectrum diagram. (**a**) Misalignment of gears. (**b**) Spectrum diagram.

**Figure 3 sensors-19-04069-f003:**
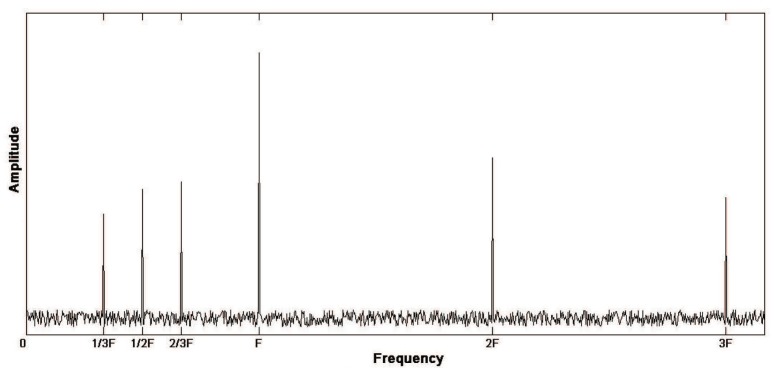
Spectrum diagram of excessive backlash of gear tooth.

**Figure 4 sensors-19-04069-f004:**
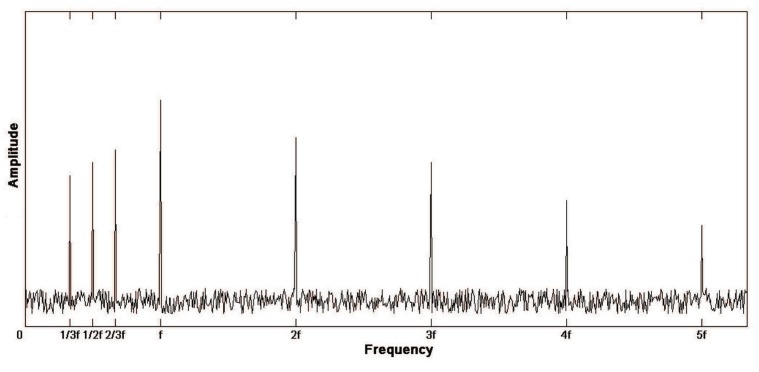
Spectrum diagram of loose mounting of bearing seat.

**Figure 5 sensors-19-04069-f005:**
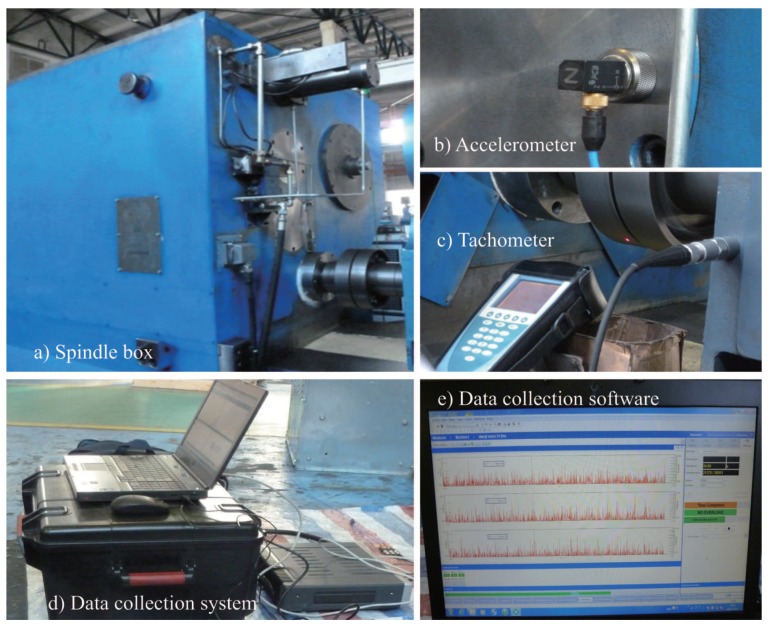
Sensors and signal acquisition equipment. (**a**) Spindle box. (**b**) Accelerometer. (**c**) Tachometer. (**d**) Data collection system. (**e**) Data collection software.

**Figure 6 sensors-19-04069-f006:**
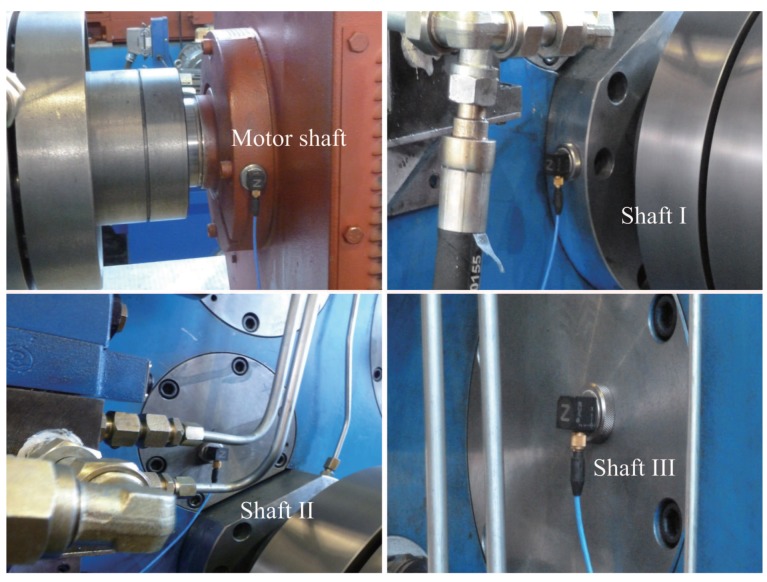
The layout of the sensor when measuring each axis.

**Figure 7 sensors-19-04069-f007:**
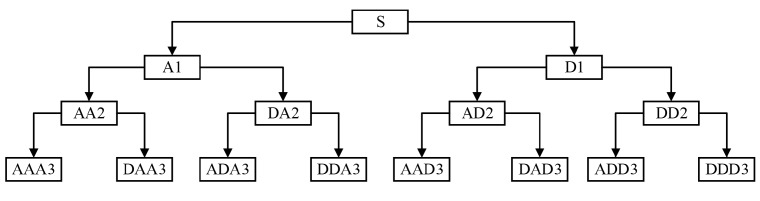
Structure diagram of three-layer wavelet packet decomposition.

**Figure 8 sensors-19-04069-f008:**
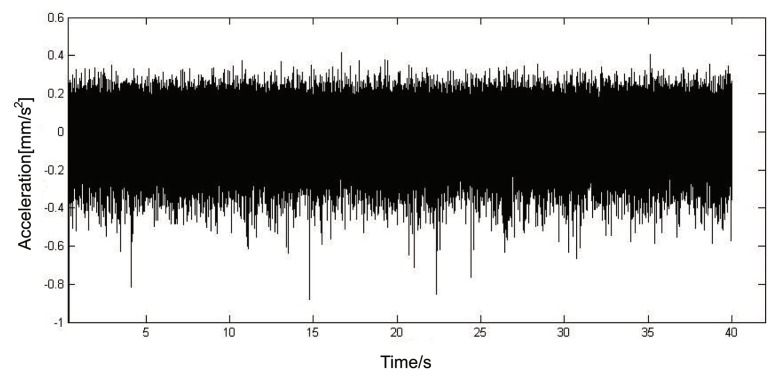
A time domain waveform on shaft II.

**Figure 9 sensors-19-04069-f009:**
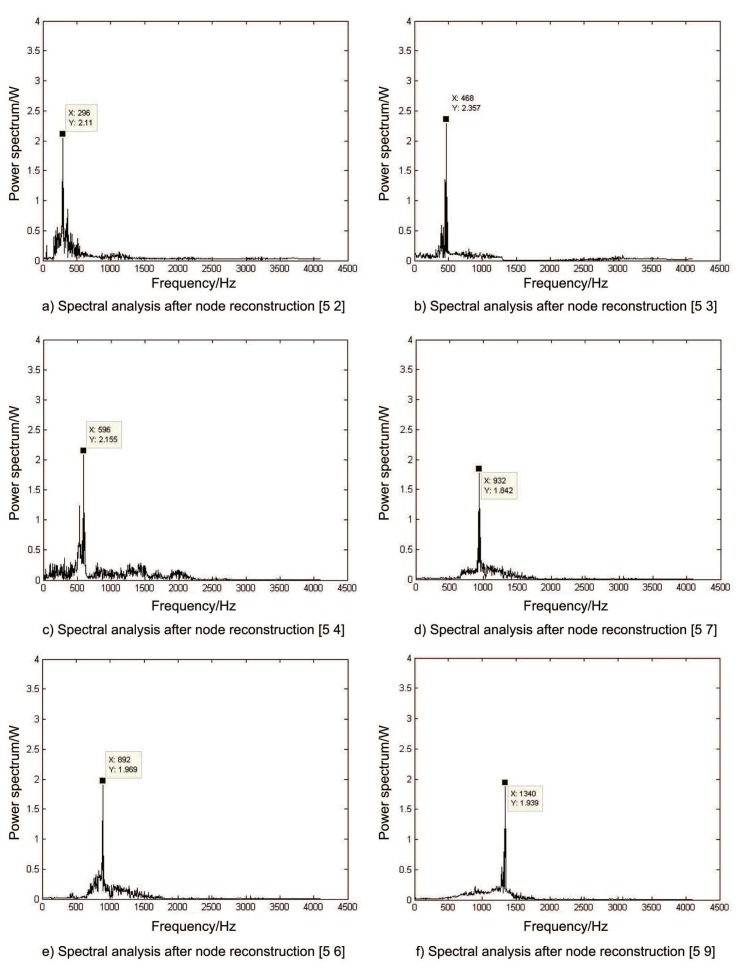
Data processing results on shaft II. (**a**) Spectral analysis after node reconstruction [5 2]. (**b**) Spectral analysis after node reconstruction [5 3]. (**c**) Spectral analysis after node reconstruction [5 4]. (**d**) Spectral analysis after node reconstruction [5 7]. (**e**) Spectral analysis after node reconstruction [5 6]. (**f**) Spectral analysis after node reconstruction [5 9].

**Figure 10 sensors-19-04069-f010:**
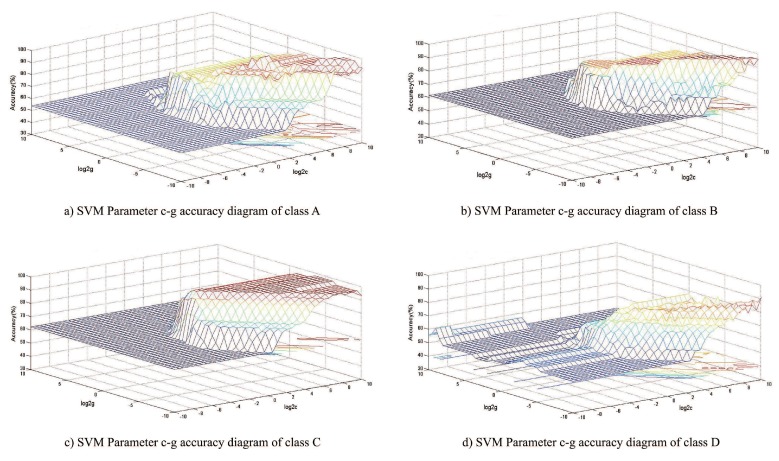
Optimization results of various SVM parameters c–g. (**a**) SVM parameter c-g accuracy diagram of class A. (**b**) SVM parameter c-g accuracy diagram of class B. (**c**) SVM parameter c-g accuracy diagram of class C. (**d**) SVM parameter c-g accuracy diagram of class D.

**Table 1 sensors-19-04069-t001:** Comparison of different transformation modes under the basis function of coif1.

No.	Discrete $E_{\mathrm{energy}}$ (J)	Wavelet $E_{\mathrm{entropy}}$	Transform E–S Ratio	Wavelet $E_{\mathrm{energy}}$ (J)	Packet $E_{\mathrm{entropy}}$	Transform E–S Ratio
1	464.077	15.145	30.642	464.099	15.126	30.683
2	230.455	15.181	15.181	230.496	15.157	15.207
3	3562.405	17.064	208.772	3562.534	16.925	210.485
4	2801.081	16.863	166.109	2081.192	16.783	166.910
5	3155.939	17.006	185.583	3156.027	16.960	186.088
6	3841.223	17.038	225.451	3841.373	16.882	227.540

**Table 2 sensors-19-04069-t002:** The results of wavelet packet decomposition based on different wavelet basis functions.

Wavelet Basis	$E_{\mathrm{energy}}$ (J)	$E_{\mathrm{entropy}}$	E–S Ratio	Wavelet Basis	$E_{\mathrm{energy}}$ (J)	$E_{\mathrm{entropy}}$	E–S Ratio
Db8	465.160	15.042	30.923	Sym6	464.450	15.037	30.886
Db20	467.251	15.013	31.124	Bior3.5	1436.305	13.892	103.391
Coif2	464.570	15.068	30.831	Bior6.8	481.790	15.003	32.114
Coif4	465.656	15.028	30.986	Rbio2.8	627.791	14.864	42.236
Sym3	464.253	15.080	30.787	Rbio5.5	601.328	14.785	40.670

**Table 3 sensors-19-04069-t003:** Summary of characteristic frequencies.

	Characteristic Frequencies
Shaft rotation frequency *f*	1/3*f*, 1/2*f*, *f*, 2*f*, 3*f*, 4*f*, 5*f*
Gear engagement frequency $F_{1}$, $F_{2}$	1/3$F_{1}$, 1/2$F_{1}$, $F_{1}$, 2$F_{1}$, 3$F_{1}$, 1/3$F_{2}$, 1/2$F_{2}$, $F_{2}$, 2$F_{2}$, 3$F_{2}$
Side frequency of mesh frequency	$F_{1}+f$, 2$F_{1}+f$, 3$F_{1}+f$, $F_{2}+f$, 2$F_{2}+f$, 3$F_{2}+f$

**Table 4 sensors-19-04069-t004:** Combinations of different fault characteristic frequencies.

The Fault Types	Combination of Characteristic Frequencies
A-Misalignment of axes *f*	*f*, 2*f*, 3*f*
B-Loose mounting of bearing seat	1/3*f*, 1/2*f*, *f*, 2*f*, 3*f*, 4*f*, 5*f*
C-Misalignment of gears	$F_{1(2)}$, $F_{1(2)}+f$, $2F_{1(2)}$, 2$F_{1(2)}+f$, 3$F_{1(2)}$, 3$F_{1(2)}+f$
D-Large side clearance of meshing teeth	1/3$F_{1(2)}$, 1/2$F_{1(2)}$, $F_{1(2)}$, 2$F_{1(2)}$, 3$F_{1(2)}$

**Table 5 sensors-19-04069-t005:** The results of parameter optimization under different SVM classifiers.

SVM Code	Optimal Parameter *c*	Optimal Parameter *g*	Cross Validation Accuracy
A	16	0.1768	96.16%
B	11.3137	0.0313	96.15%
C	0.3536	0.2500	97.78%
D	1024	0.0098	91.11%

**Table 6 sensors-19-04069-t006:** Summary of classification results of test samples.

SVM Code	Test Sample Size	Correct Number	Test Accuracy
A	22	18	81.81%
B	22	21	95.45%
C	30	28	93.33%
D	30	28	93.35%

**Table 7 sensors-19-04069-t007:** Test results of different research works using SVM.

	Test Sample Size	Correct Number	Test Accuracy
This research (SVM-D)	60	58	96.67%
Wu’s research (SVM) [22]	60	57	95%
Jiang’s research (SVM) [23]	60	56	93.33%

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
