# Peer review of "Fault Diagnosis of Rotary Parts of a Heavy-Duty Horizontal Lathe Based on Wavelet Packet Transform and Support Vector Machine"

_sensors, 2019, doi:10.3390/s19194069_

Round 1
Reviewer 1 Report
I suggest not using an acronym in the title; Citations are missing in important parts of this paper, e.g., “fast Fourier transform”, “Wigner-Ville distribution, ”artificial neural networks”, “hidden Markov model”, “genetic algorithm”, “Shannon entropy”, “Coiflet wavelet” I observe the total absence of citation in the equations; I recommend using citations from benchmark publications, for example, in relation to SVM [A], wavelet transform [B], etc.; Can Bibliography (15 publications) be considered the state of the art in the subject matter of this paper?; Bibliography Review. For example, the correct name of one of the authors of the reference [12] is “Sadegui”; I suggest to the Authors to highlight (objective and clearly) the innovation of this proposal in relation to literature.
References
[A] Vapnik, V. “The nature of statistical learning theory”, Springer-Verlag, New York, 1995.
[B] Daubechies, I. “Ten Lectures on Wavelets”, Society for Industrial and Applied Mathematics-SIAM, Philadelphia, Pennsylvania, , 1992.
Reviewer 2 Report
In this paper, Jin et al., developed SVM-based method to classify the faulty types. I found the manuscript interesting and the proposed method promising. I would like to recommend it for publication subject to the authors’ revision based on my comments.
1. Provide scripts and needed data so that the work can be reproduced by readers. Don't forget a license for the code/scripts.
2. The current version of the research article seems review. I suggest the author to follow the journal format (research article), such as, introduction, materials and methods, result and discussion and conclusions.
3. Provide definition for evaluation metrics.
4. Generally, external validation is needed to evaluate the developed prediction model. In this regard, I suggest authors use only 80% of the data for the development of prediction model and the remaining 20% for external or independent validation. This will tell how much the developed prediction model transferable to the unknown data.
5. SVM has been widely applied in the field of computational biology. I suggest the authors to mention this point in page 13 and line 392 with the following references (PMID: 29100375 , 31099381, 31146255, 31013619, 28419290).
6. Change figure 10 to 2X2, it will be more easy to compare.
Reviewer 3 Report
The authors propose a methodology to diagnose faults in a gear box using WPT, this reviewer has the next comments:
Can you show the angle alpha in figure 1.a?
How do you get spectrum of figure 1.b?
It will be good add the model equation of the tested faults
The same for figure 2 and 3 as figure 1, please add labels to explain the figure, and add the model equations of these faults to understand how did you get the spectrums?
Can you show the frequencies in Table 4 in the figures?
Please add a comparative table with others similar works to know the effectiveness of you method.
Round 2
Reviewer 1 Report
-
Reviewer 2 Report
The authors have addressed all my comments. Therefore, I recommend this paper for publication.